# Glutamate Receptor Dysregulation and Platelet Glutamate Dynamics in Alzheimer’s and Parkinson’s Diseases: Insights into Current Medications

**DOI:** 10.3390/biom13111609

**Published:** 2023-11-03

**Authors:** Deepa Gautam, Ulhas P. Naik, Meghna U. Naik, Santosh K. Yadav, Rameshwar Nath Chaurasia, Debabrata Dash

**Affiliations:** 1Center for Advanced Research on Platelet Signaling and Thrombosis Biology, Department of Biochemistry, Institute of Medical Sciences, Banaras Hindu University, Varanasi 221005, India; 2The Cardeza Foundation for Hematologic Research, Center for Hemostasis, Thrombosis and Vascular Biology, Department of Medicine, Thomas Jefferson University, Philadelphia, PA 19107, USA; ulhas.naik@jefferson.edu (U.P.N.); meghna.naik@jefferson.edu (M.U.N.); santosh.yadav@jefferson.edu (S.K.Y.); 3The Department of Neurology, Institute of Medical Sciences, Banaras Hindu University, Varanasi 221005, India; goforrameshwar@gmail.com

**Keywords:** Alzheimer’s disease, Parkinson’s disease, platelet, glutamate, NMDA receptor, AMPA receptor, kainate receptor, metabotropic glutamate receptor

## Abstract

Two of the most prevalent neurodegenerative disorders (NDDs), Alzheimer’s disease (AD) and Parkinson’s disease (PD), present significant challenges to healthcare systems worldwide. While the etiologies of AD and PD differ, both diseases share commonalities in synaptic dysfunction, thereby focusing attention on the role of neurotransmitters. The possible functions that platelets may play in neurodegenerative illnesses including PD and AD are becoming more acknowledged. In AD, platelets have been investigated for their ability to generate amyloid-ß (Aß) peptides, contributing to the formation of neurotoxic plaques. Moreover, platelets are considered biomarkers for early AD diagnosis. In PD, platelets have been studied for their involvement in oxidative stress and mitochondrial dysfunction, which are key factors in the disease’s pathogenesis. Emerging research shows that platelets, which release glutamate upon activation, also play a role in these disorders. Decreased glutamate uptake in platelets has been observed in Alzheimer’s and Parkinson’s patients, pointing to a systemic dysfunction in glutamate handling. This paper aims to elucidate the critical role that glutamate receptors play in the pathophysiology of both AD and PD. Utilizing data from clinical trials, animal models, and cellular studies, we reviewed how glutamate receptors dysfunction contributes to neurodegenerative (ND) processes such as excitotoxicity, synaptic loss, and cognitive impairment. The paper also reviews all current medications including glutamate receptor antagonists for AD and PD, highlighting their mode of action and limitations. A deeper understanding of glutamate receptor involvement including its systemic regulation by platelets could open new avenues for more effective treatments, potentially slowing disease progression and improving patient outcomes.

## 1. Introduction

The restricted equilibrium between excitatory and inhibitory neurotransmission drives the sophisticated neuronal networks of the brain, which control everything from memory to motor control [1]. This equilibrium is crucially mediated by glutamate, the primary excitatory neurotransmitter in the central nervous system, which also mediates synaptic plasticity, memory formation, and learning. These activities rely heavily on glutamate receptors, and disruption in glutamatergic signaling has long been linked to several neurological and mental disorders [2,3].

Two of the most prevalent NDDs in the world, AD and PD, have both been connected to changes in glutamatergic neurotransmission. AD and PD are both neurodegenerative diseases, but they primarily affect different regions of the brain and manifest distinct clinical symptoms. Despite these differences, at the synaptic level, both diseases demonstrate pathophysiological alterations, particularly in relation to glutamate receptors [4,5,6,7,8]. Glutamate excitotoxicity is a complex biological process that initiates when glutamate receptors are activated, leading to damage in dendrites and eventually causing cell death [9]. This mechanism impacts various cellular compartments including the cytosol, mitochondria, endoplasmic reticulum, and the nucleus [10] (Figure 1). Interestingly, even normal activation levels of glutamate receptors can harm neurons, particularly in situations involving metabolic and oxidative stress, such as after a stroke, during traumatic brain injury, or in age-related ND conditions like AD and PD. The process is set off by an excessive flow of sodium (Na^+^) and calcium (Ca^2+^) ions through the cell membrane, which is a result of the opening of glutamate receptor channels. Various factors, such as the type of neuron, its developmental stage, and environmental influences, can affect the severity of excitotoxicity. The neuron’s ability to remove and buffer Ca^2+^ ions, facilitated by mechanisms like Ca^2+^-ATPases and calcium-binding proteins, also plays a critical role in determining its vulnerability to excitotoxicity [9,11]

This review aims to delve into the current understanding of the role of glutamate receptors in the context of AD and PD, exploring their potential as therapeutic targets and their implications in the broader landscape of ND research. This review also briefly discusses the role of platelets in the release and uptake of glutamate and its significance in these ND conditions.

## 2. Alzheimer’s Disease

Numerous conditions or traumas that both directly and indirectly harm the brain can result in dementia [12,13]. The most prevalent type of dementia is AD, which accounts for 60–70% of cases. It is a progressive NDD beginning with mild memory loss and possibly with time can evolve to more severe impairment like loss of the ability to carry on a conversation and respond to the environment [14,15]. It is characterized by neuritic plaques and neurofibrillary tangles (NFTs) particularly in the medial temporal lobe and neocortical regions of the brain (Figure 2A). These can impair basic functions of mind, memory, and language, such as the capacity to speak clearly and react appropriately to one’s environment [16,17]. According to the World Health Organization (WHO) report, more than 55 million individuals worldwide currently suffer from dementia, and more than 60% of them reside in a low- and middle-income nations. It is becoming an urgent concern with about 10 million new cases per year. Dementia, which is currently the seventh most common cause of mortality in the world, is a significant cause of dependency and disability among the aging population. The picture is made more difficult by the gender dynamics of dementia.

There are two distinct neuropathological alterations associated with AD, i.e., positive, and negative lesions (Figure 2B). NFTs, amyloid plaques, dystrophic neurites, neuropil threads, and other deposits that impair brain function are the hallmarks of positive lesions. Negative lesions are caused by a loss of neurons, neuropil, and synapses, which causes severe atrophy or shrinking in certain brain regions and cognitive deficiencies. In addition to these lesions, chronic neuroinflammation, oxidative stress, and damage to cholinergic neurons which are essential for memory and learning are other variables that contribute to neurodegeneration in AD [16,17,18,19].

AD is characterized by senile plaques, which are extracellular beta-amyloid protein (Aβ) deposits that can take on a variety of shapes, including neuritic, diffuse, dense-cored, classic, and compact type plaques [20]. The transmembrane amyloid precursor protein (APP), which forms these plaques, is broken down by proteolytic enzymes such as β-secretase and γ-secretase into various amino acid fragments, including Aβ-40 and Aβ-42 [21,22]. The accumulation of Aβ can result in the creation of massive, insoluble amyloid fibrils that aid in the formation of plaques as well as soluble oligomers that spread throughout the brain [23]. These accumulates cause neurotoxicity and compromised brain function, especially in the hippocampus, amygdala, and cerebral cortex. Axons and dendrites are damaged, synapses are lost, and later cognitive deficits result from such intense plaque formation, which activates astrocytes and microglia. The activation of these cells suggests an inflammatory response, which may contribute to additional neuronal damage and, subsequently, cognitive deficits [17,20].

AD is also characterized by aberrant brain structures called NFTs, which are mostly made of hyperphosphorylated tau protein [18,24]. These filaments can twist and create paired helical filaments, which build up in different brain regions and cause the loss of critical proteins and microtubules. These tangles interfere with the cytoskeleton’s regular operation, which results in the loss of vital microtubules and the related proteins. Because of this disruption, neurons are unable to transfer nutrition and other essential molecules, which ultimately results in neuronal death [18,24].

The complexity of AD is shown by the connections between the hereditary and environmental factors. Age is still the biggest risk factor for AD, and beyond the age of 65, there is a higher chance of it happening. AD has been classified into early-onset Alzheimer’s disease (EOAD) and late-onset Alzheimer’s disease (LOAD) based on age at which the disease manifests and whether it is inherited. EOAD starts before the age of 65 and it follows Mendelian inheritance, whereas LOAD occurs after the age of 65, and it does not show any familial link. EOAD is caused due to the mutation in the gene like APP, Presenilin 1 (PSEN1) and Presenilin 2 (PSEN2) (these genes encode γ-secretase: an enzyme complex involved in APP processing) [25,26]. The fact that variations in other genes like apolipoprotein E (APOE), triggering receptor expressed on myeloid cells 2 (TREM2), Clusterin (CLU), and several others are also linked to an increased risk of AD emphasizes the genetic complexity of the condition [27,28]. The trisomy of chromosome 21, where the APP gene is located, also increases the risk of dementia in people with Down syndrome [29]. Numerous environmental elements in addition to genetic variables are very important. Chronic disorders including cardiovascular diseases and type 2 diabetes are also known to increase the risk [30,31]. Cognitive impairment has been linked to inflammation, which results from injury or infection [32]. The complexity of AD is heightened by additional environmental risk factors such stress, pollution, and heavy metal exposure [15,32,33].

## 3. Parkinson’s Disease

PD is a progressive neurological disorder resulting from brain degeneration. It is notably prevalent, ranking as the second most common age-related neurodegenerative condition [34]. This disorder not only affects muscular control, balance, and movement; it can also have a wide range of additional consequences on your senses, thinking capacity, mental health, and more [34]. Parkinson’s symptoms are broadly classified into motor and non-motor symptoms. Motor symptoms include tremor, slowed movement (bradykinesia), rigid muscles, impaired posture/balance, loss of automatic movements (like eye blinking), trouble swallowing (dysphagia) and change in speech (hypophonia) and writing, whereas non-motor symptoms include cognitive impairment, mental health disorders, dementia, sleep disorders pain, and sensory symptoms [35].

As per the WHO, the number of patients with PD has doubled over the previous 25 years. Over 8.5 million people had PD in the world as of 2019. Their lives are affected by this illness, which also causes disability. In addition, 329,000 people died from PD in 2019, which is more than a twofold increase since 2000. This demonstrates how widespread PD has grown to be.

Another most important hallmark of PD is the presence of Lewy bodies (protein clumps), which are composed of a misfolded and aggregated protein called α-synuclein. The accumulation of the Lewy bodies causes the death of dopamine-producing neurons, leading to classical motor symptoms in Parkinson’s [36,37]. Non-genetic factors also lead to PD. It has been reported that long exposure to pesticides and herbicides can increase the risk of PD [38]. Age is another notable factor, although early onset can manifest before age 50. Gender also plays a role, as men are more prone to develop Parkinson’s compared to women [39,40]. Oxidative stress and mitochondrial dysfunction have also been suggested as contributing factors as they lead to cell death [41,42].

## 4. Glutamate Receptors: Key Players in Neurotransmission, Learning and Brain Health

Glutamate is an important neurotransmitter in the central nervous system (CNS), where it plays an important role in cognitive functions such as learning, memory, and neuronal communication. It is an excitatory neurotransmitter that stimulates and activates neurons [43,44]. Glutamate acts on a wide range of receptors that are a varied set of proteins found in the CNS and are essential in a variety of physiological activities, including synaptic transmission, learning, memory, and neural plasticity regulation [43,45]. Glutamate receptors can be broadly classified into two main categories: ionotropic receptors and metabotropic receptors (Figure 3) [46].

### 4.1. Ionotropic Receptors

Ionotropic glutamate receptors are a class of ligand-gated ion channels that play a critical role in mediating excitatory neurotransmission in the brain. These receptors are activated by the neurotransmitter glutamate, which is the primary excitatory neurotransmitter in the mammalian central nervous system. Upon activation, ionotropic glutamate receptors allow the flow of ions like Na^+^, K^+^, and Ca^2+^ across the neuronal membrane, leading to cellular depolarization and subsequent propagation of the neural signal. The ionotropic glutamate receptor is subdivided into three main types—N-methyl-D-aspartate (NMDA), alpha-amino-3-hydroxy-5-methyl-4-isoxazolepropionic acid (AMPA), and kainate receptors—each with unique pharmacological properties and roles in synaptic plasticity, learning, and memory [46,47].

#### 4.1.1. N-Methyl-D-aspartate Receptor (NMDAR)

The NMDAR, a subtype of the ionotropic glutamate receptor, was discovered in the 1980s and has since become one of the most thoroughly studied receptors in neuroscience [48]. It plays an important role in synaptic plasticity, learning, memory formation, and other physiological processes in the CNS [49]. Long-term potentiation (LTP), a mechanism critical for learning and memory, is mediated by the NMDAR [50]. The NMDAR is a heteromeric protein complex made up of four subunits that are organized around a core ion channel [51]. The primary subunits that form the NMDAR are glutamate-binding NR2 subunits and glycine-binding NR1 subunits. The NR2-glutamate binding subunits are usually represented as NR2A, NR2B, NR2C and NR2D [52]. The NMDAR’s functional characteristics may be influenced by the NR2 subunit that is present. For instance, receptors with NR2A or NR2B subunits are frequently located at synaptic locations and contribute to synaptic plasticity; in contrast, receptors with NR2C or NR2D subunits are extrasynaptic and may contribute to neurotoxicity [53]. The NR1 glycine-binding subunits are necessary for the receptor to function because they bind the co-agonists like glycine or D-serine [52]. There is only one kind of NR1 subunit, but it can go through post-translational changes and alternative splicing to create many isoforms with marginally different characteristics [54]. The binding of glutamate as well as the co-agonists glycine or D-serine is required for NMDAR activation [55]. A functional receptor can also be created by joining NR1 and NR2 subunits with a third type of subunit, NR3 (with subtypes NR3A and NR3B). These NR3 subunits are less well known, although they seem to change the properties of the receptor when they are integrated, for example by decreasing the receptor’s Ca^2+^ permeability [56]. The NMDAR is typically made up of two NR1 and two NR2 subunits. The precise configuration and subunit types of NR2 can have a big impact on the receptor’s conductance, sensitivity to medicines, and how quickly it desensitizes and subsequently resensitizes to glutamate. Each subunit of these receptors is structured with four main components: an amino terminal domain (ATD), a ligand-binding domain (LBD), a transmembrane domain (TMD), and a carboxyl terminal domain (CTD) [54]. These subunits assemble to form the receptor, which has three critical domains: one for capturing neurotransmitters like glutamate and glycine in the extracellular space, another that establishes the ion channel through the cell membrane, and a third that interfaces with the cell’s internal signaling pathways [57]. Together, these components make the NMDAR essential for neural signaling and synaptic plasticity. The NMDAR is voltage-dependent, i.e., its activation depends not only on the binding of glutamate but also on the electrical state of the neuron. Specifically, the ion channel of the NMDAR is blocked by magnesium ions (Mg^2+^) at the neuron’s resting potential. When the neuron is depolarized, this Mg^2+^ block is relieved, and if glutamate is bound, the transmembrane domain creates the ion channel that lets ions flow through to detect concurrent pre- and postsynaptic activity. In addition to K^+^ and Na^+^, the NMDAR ion channel has a special permeability to Ca^2+^ (Figure 4A). The cell uses Ca^2+^ as crucial second messengers to start operations like gene transcription and synaptic plasticity. The NMDAR performs the function of a “coincidence detector” due to its voltage dependence and requirement for two different types of ligands, i.e., glutamate and either glycine or D-serine. Only in the presence of both ligands and when the post-synaptic cell is sufficiently depolarized will it allow current to flow [46,47,58].

##### Role of NMDA in Alzheimer’s Disease

Although the NMDAR is essential for healthy brain function [49], AD is characterized by the dysregulation and overactivation of this receptor, which results in neuronal death and cognitive loss [59]. Several presumptions have been put forward to explain the cause of AD neuropathology like reactive oxygen species (ROS)-mediated damage [60], Aβ and Tau protein mediated toxicity, to cholinergic dysfunction, but the exact origins of AD are still unknown. However, one of the prevailing theories posits that dysregulation of the NMDAR is one of the causes of AD and therefore, the NMDAR plays a complex and multifaceted role in AD [61]. It is well known that Aβ oligomers overactivate NMDARs, which causes a significant inflow of Ca^2+^ ions into the cell [62]. Even while Ca^2+^ ions are necessary for many intracellular signaling processes, accumulating too much of them can be detrimental. Elevated intracellular Ca^2+^ can activate enzymes including phospholipases, endonucleases, and proteases that can damage DNA, the cytoskeleton, and cell membranes. Additionally, it may enhance the generation of ROS, which can cause oxidative stress and further harm to the cell. This process is called excitotoxicity, which is a process that can result in cell death and is regarded to be one of the main mechanisms behind the loss of neurons in AD [63,64]. Aβ has an impact on various NMDAR subunits as well as amounts of mRNA and protein. There are reports of decreased levels of GluN1 mRNA and GluN1 protein in AD patients, although some studies found unchanged levels of GluN1 mRNA [65]. Evidence suggests that GluN2A and GluN2B mRNA levels have reduced particularly in the entorhinal cortex and hippocampal regions of the brain, which are crucial for memory functions and are severely damaged in AD. Levels of GluN2A and GluN2B protein were also found to be decreased in the AD brain. According to reports, GluN2C and GluN2D mRNA levels were similar in AD patients and controls, indicating that these subunits may not be severely altered by AD. While GluN2A’s mRNA expression and protein levels in the hippocampus were found to be unchanged in certain investigations, GluN1 and GluN2B’s mRNA expression and protein levels were found to be drastically reduced. While evidence of lower NMDAR subunit levels—both in terms of mRNA and protein—exists, the precise subunits affected and the severity of these effects vary between research [57]. The function of the receptor may be impacted by these changes in NMDAR subunit levels, which may also contribute to the synaptic dysfunction and cognitive deficiencies found in AD. According to one study, Aβ1-42 administration causes synaptic NMDARs to internalize and NMDAR-mediated currents to be suppressed in cultured cortical neurons, resulting in suppressed synaptic glutamatergic transmission and an inhibition of synaptic plasticity [66,67].

The early synaptic alterations in AD have been examined in the study with a particular emphasis on the interaction between Aβ and the NMDAR, specifically involving an NR2B subunit. A key actor that has been demonstrated to increase the number of functioning NMDARs at the cell surface and improve channel activity is integrin β1 (receptor for Aβ oligomer) and protein kinase C (PKC). Aβ oligomers activate integrin β1 and protein kinase C, which in turn phosphorylates NR2B at serine 1303 and increases the conductance of the NMDAR, leading to a boost in Ca^2+^ influx and excitotoxicity. The pathway emphasizes the integrin β1/PKC/NMDAR signaling axis as an early AD event (Figure 5) [57].

Like Aβ, tau is a kind of protein that is thought to contribute to the pathophysiology of AD [68,69]. It is mostly found in neurons, where it stabilizes microtubules, a component of the cell’s cytoskeleton, to maintain the structural integrity of neurons under normal conditions. NFTs, which are abnormally hyperphosphorylated tau proteins that have begun to cluster inside neurons, are a symptom of AD. These tangles obstruct the neurons’ ability to function normally and finally cause neuronal death [69]. Synapses, which are the connections between neurons, might become dysfunctional due to hyperphosphorylated tau. Any damage to synapses could affect NMDAR activity because NMDARs are essential for synaptic function [49]. A tyrosine kinase called Fyn, which phosphorylates NMDARs, has been demonstrated to interact with tau. Tau appears to function as a scaffold that draws Fyn and the NMDAR together into proximity, promoting the phosphorylation of the receptor (Figure 5) [70,71]. Studies on tau knockout mice have revealed that their NMDAR function is diminished. This shows that tau contributes to the maintenance of NMDAR activity in its natural state, and it may be a factor in the deregulation of NMDARs observed in AD [72].

##### Role of NMDAR in Parkinson’s Disease

Many NDDs, including PD, are thought to have the characteristic trait of glutamate-induced excitotoxicity, which has been connected to changes in the expression of glutamate transporters and receptors presumably in connection with inflammatory processes [73,74]. The motor and non-motor symptoms of PD, as well as the dyskinesia brought on by L-3,4-dihydroxyphenylalanine (L-DOPA) therapy, are thought to be related to the failure of NMDAR transmission. Both dopamine depletion and L-DOPA therapy in PD alter the NMDAR, specifically the GluN2A, GluN2B, and GluN2D subunits. Both PD patients and several animal models have shown these changes [75]. It has been observed that there is an increase in the GluN2A/GluN2B subunit ratio and increased NMDA-sensitive glutamate binding in the affected brain areas, which increases the Ca^2+^- mediated signal transduction pathway that accelerates the degenerative process (Figure 1) [76]. One of the studies has explored the complex mechanisms underlying Levodopa-Induced Dyskinesias (LIDs), which is a common yet debilitating side effect experienced by PD patients undergoing long-term levodopa therapy. While glutamate, the brain’s principal excitatory neurotransmitter, has been implicated in animal models, the direct evidence in humans remains scant. The research employs 11C-CNS 5161 PET scans, a specialized form of positron emission tomography, to probe the activity of the NMDAR, which is a specific subtype of the glutamate receptor. CNS 5161 acts as a non-competitive antagonist for these receptors, serving as a potent tracer to identify activated ion channels. Patients participating in the study underwent two scans—once when they are “OFF” levodopa medication, having had all Parkinson’s medications withdrawn for 12 h, and once when they are “ON” levodopa, administered one hour before the scan. This dual-scan approach allowed for a comparison of NMDAR activity under both conditions. The study’s hypothesis posits that altered glutamatergic transmission, evidenced by changes in NMDAR activation, may be associated with the development and persistence of LIDs [77]. NMDA antagonists have been shown to significantly protect dopaminergic neurons in the substantia nigra, raising the possibility that they could not only manage symptoms but also slow disease progression [78,79]

PD affects NMDARs in the brain, specifically within the striatal neurons. It was analyzed that how individual subunits of the NMDARs were affected in two distinct cellular locations: the surface of the neuron and inside the neuron, focusing primarily on three NMDA subunits: GluN1, GluN2A, and GluN2B. It was discovered that when the dopaminergic pathway was damaged by the neurotoxin, there was a significant increase in the GluN1 subunits on the surface of the striatal neurons. In contrast, the intracellular levels of GluN1 were reduced. For another subunit, GluN2B, they found an increase in its abundance on the surface of neurons in the damaged area. The levels of GluN2B inside the cell remained stable. Meanwhile, the GluN2A subunit levels did not show significant changes either on the cell surface or within the cell. These findings suggest that the brain might be trying to compensate for the loss of dopamine by altering the expression of these NMDAR subunits. Particularly, the increase in surface expression of GluN1 and GluN2B could be an adaptive response to the altered neuronal environment caused by the dopamine depletion in PD [80].

A recent study has explored the role of NMDAR antibodies in PD, particularly focusing on their impact on cognitive and neuropsychiatric symptoms. Blood samples from PD patients and healthy controls were examined for the presence of different types of NMDAR antibodies (IgA, IgM, and IgG) as well as other neurological biomarkers like phosphorylated tau 181 (p-tau181) and NFTs. The cognitive and neuropsychiatric states of the participants were also assessed through established tests like the Mini-Mental State Examination (MMSE). The study found that 9% of PD patients had NMDAR antibodies, particularly IgA type, compared to only 3% in the control group. These antibodies were not linked to age, gender, or disease duration but were associated with a more rapid cognitive decline over time as measured by the MMSE. This suggests that NMDAR antibodies could be a significant factor in cognitive decline in Parkinson’s patients [81].

#### 4.1.2. α-Amino-3-hydroxy-5-methyl-4-isoxazolepropionic Acid Receptor (AMPAR)

AMPAR is an essential ionotropic glutamate receptor that plays a pivotal role in fast excitatory synaptic transmission within the mammalian CNS. Functioning as a molecular gateway for Na^+^ and, occasionally, Ca^2+^, AMPARs are integral to the initiation of cellular depolarization and neural signaling. These receptors are especially critical for cognitive functions like learning and memory and have been implicated in various neurological and psychiatric conditions [82,83,84].

The AMPAR is a tetramer, which means it consists of four subunits. Four distinct genes each encode one of the four subunit types: GluA1, GluA2, GluA3, and GluA4 (Figure 4B). To create an active receptor, the subunits can come together in a variety of ways, and each subunit arrangement gives the receptor a unique set of characteristics [84,85]. Each subunit of AMPAR is comprised of an extracellular domain, transmembrane domain, and intracellular domain. The extracellular domain consists of two component segments, i.e., the N-terminal domain (NTD) and the ligand-binding domain (LBD) [46]. The NTD is involved in receptor modulation and subunit assembly. AMPARs are activated by the neurotransmitter glutamate, which binds to the receptor in the LBD. The transmembrane domain is made up of three transmembrane segments (M1, M3, and M4) and a re-entrant loop (M2). A conformational shift brought on by glutamate’s binding to the LBD causes the ion channel in the transmembrane domain to open, allowing ions to pass through the receptor. The intracellular domain or the C-terminal domain structure is not constant and fluctuates substantially across various subunits. The postsynaptic density is a specific region of the cell where signaling molecules are concentrated. The postsynaptic density is where the CTD plays a function in trafficking and anchoring the receptor. The overall design of the AMPAR enables it to fulfill its role as a glutamate receptor, driving rapid synaptic transmission in the brain. When glutamate binds to the LBD, it causes the ion channel to open, letting cations—mainly Na^+^ and, in some circumstances, Ca^2+^—into the cell [46,84]. This causes the cell to become depolarized and transmits the electrical signal. Because of their quick opening and closing in response to glutamate binding, AMPARs are essential for synaptic plasticity, which is a critical step in learning and memory. Synapses can become stronger or weaker over time. The variety in AMPAR function across various brain areas and neuronal populations is made possible by the variable subunit composition [46].

##### Role of AMPAR in Alzheimer’s Disease

The dysregulation of excitatory glutamate receptors, especially AMPAR, has been shown to impede LTP and impair synaptic transmission, which can result in deficiencies in learning and memory. This is a defining characteristic of AD etiology and takes place before symptomatic plaques and neuronal loss are found [86,87]. Different AD models have shown a dysregulation of AMPAR abundance, trafficking, localization, and function [88]. The GluA1 and GluA2 subunits are the most thoroughly researched AMPAR subunit. The consistent AMPAR activity during rest can help produce and release Aβ into the interstitial fluid, which is the fluid that fills the gaps between cells. This shows that the generation and release of Aβ can be regulated by regular glutamatergic transmission, which involves AMPARs [89]. A reduction in the levels of Aβ may occur when AMPAR signaling is triggered or activated. Through NMDARs, for instance, this may happen. To unlock NMDARs from their magnesium barrier and enable their activation, AMPARs must be stimulated to depolarize the neuron. Then, intracellular signaling pathways can be activated by the activation of NMDARs to either increase or decrease the production of Aβ. However, irrespective of NMDARs, induced AMPAR signaling can also lower Aβ levels. One such method entails greater extracellular Aβ clearance at least in part because of improved signaling by the cytokine interleukin-6 (IL-6), which is a component of immunological responses [89].

It is generally accepted that AD starts as a malfunction of synapses and progresses to dementia and cognitive decline. Although it has not been tested experimentally, homeostatic synaptic scaling (homeostatic form of synaptic plasticity that adjusts the strength of all neuron’s excitatory synapses up or down to stabilize neural firing rates) is a mechanism that could be significant in the start of AD. Recent studies have demonstrated that the development of AD is accompanied by a selective decline in AMPARs, which has profound physiological and cognitive ramifications. Synaptic scaling, a process that can be implicated in the illness process in vivo, is assumed to be the cause of the decrease in AMPARs. Scaling-based computational associative memory research can accurately simulate the typical pattern of amnesia in AD. Thus, the sickness may be caused by homeostatic synaptic scaling [90]. This was also confirmed using double knockin mice with human mutations in the PSEN1 and APP genes (these genes are particularly important since early-onset familial AD [25] can mimic AD). This study found that these mice have an age-related downscaling of spontaneous, tiny currents and AMPAR-mediated evoked currents at the synaptic level. Additionally, these double knockin mice demonstrated compromised bidirectional synaptic plasticity, affecting both long-term potentiation and long-term depression, along with reduced memory flexibility. These findings shed light on the critical role AMPARs may play in the development and progression of AD, suggesting that the downscaling of postsynaptic AMPAR function could be a key factor in the cognitive deficits observed in the disease [90].

The hippocampus, a part of the brain crucial for learning and memory, experiences changes in the expression and distribution of AMPAR in AD. Particularly, hippocampal interneurons selectively receive less AMPA current, which reduces network oscillatory activity [91]. Aβ levels that are too high that can impair excitatory synaptic transmission and plasticity, mostly because the brain’s AMPAR is dysregulated. Aβ oligomers can attach to AMPARs, causing an internalization and degradation of AMPAR, because of which synaptic transmission and plasticity decrease. Age influences the transition of AMPARs from synapses to intracellular compartments in the CA1 hippocampus region of APP/PS1 transgenic mice. Particularly, 12-month-old mice have a substantial increase in intracellular AMPAR clusters as compared to 3-month-old animals. In this mouse model of AD, this suggests that the transition of AMPARs from synapses to intracellular compartments is a gradual process that happens with age. The cognitive processes that are most severely affected in AD are synaptic transmission and plasticity, and Aβ-induced dysregulation of AMPAR trafficking can result in a decrease in these processes. In addition, the Aβ-induced internalization of AMPARs might result in synaptic loss, one of the first processes in AD pathogenesis, before neuronal death [92]. To create effective treatment plans for AD, it is essential to comprehend the molecular mechanisms behind the failure of AMPAR trafficking caused by Aβ.

The importance of the GluA2 subunit AMPAR was identified using RNA editing modifications in a hAPP-J20 animal model of AD. This model mimics many symptoms of AD, including neurodegeneration, inflammation, Aβ-plaque production, and cognitive impairments. The influx Ca^2+^ is tightly regulated by the GluA2 subunit of AMPARs. A glutamine in the mRNA of the GluA2 subunit is specifically changed to an arginine by the enzyme ADAR2 (adenosine deaminase acting on RNA), making the AMPAR Ca^2+^-impermeable. This is significant because uncontrolled Ca^2+^ influx can result in excitotoxicity, which can harm or kill neurons. A mouse model with more naturally occurring GluA2 was produced to investigate this phenomenon further. This model demonstrated neurodegeneration, elevated inflammation, and alterations in dendritic length and spine density, which is consistent with the idea that GluA2 RNA editing is essential for neuroprotection in the hippocampal CA1 region, increased spine density, and improved performance in memory task [93,94].

##### Role of AMPAR in Parkinson’s Disease

AMPARs are essential for controlling glutamate activity, which is thought to be one of the contributing factors in PD. Movement is impacted by PD, and the popular drug levodopa occasionally results in dyskinesia, or involuntary jerky movements. In a study using monkeys with a condition resembling PD and focusing on the striatum, a region of the brain, and AMPARs, it was discovered that areas of the striatum with higher AMPARs were developing dyskinesia. Levodopa avoided this rise when it was taken with another drug like CI-1041 (NMDA-antagonist) and cabergoline (D2 receptor agonist) [95]. In other words, the research raises the possibility that levodopa-induced jerky movements in Parkinson’s patients are mediated by AMPAR [96]. According to the research, the motor problems that PD patients suffer after taking levodopa may possibly be related to alterations in AMPAR sensitivity in particular parts of the brain [97]. It was discovered that the lateral putamen binding of (3)H-AMPA was higher (+23%) in PD patients with motor problems. Contrarily, this binding was 16% less in the caudate nucleus of PD patients than in controls. PD is often accompanied by anxiety disorders. Anxiety in PD is modulated by AMPARs, which are found in the lateral habenula (LHb): a part of the brain that affects the monoaminergic system. Researchers investigated how neurotransmitter levels and anxiety-like behaviors in PD and control rats were affected by the AMPAR agonist (S)-AMPA and antagonist NBQX and it was found that in contrast to NBQX, (S)-AMPA had a calming effect while raising dopamine and serotonin levels in the basolateral amygdala [98].

Another study has suggested that the movement problems associated with nigrostriatal dopaminergic neurodegeneration may be caused by Ca^2+^- and Zn^2+^-permeable GluR2-lacking AMPAR. AMPA injection into the substantia nigra pars compacta (SNc) resulted in an increase in turning behavior and a loss of nigrostriatal dopaminergic neurons, which is indicative of movement problems. Since the movement disturbance and the resulting neurodegeneration were both inhibited by the co-injection of Zn^2+^ chelators (ZnAF-2DA and TPEN), it was shown that these effects were caused by intracellular Zn^2+^ dysregulation. The study also showed that TPEN prevented the rapid rise in nigral intracellular Zn^2+^ levels that was caused by AMPA, indicating that AMPA plays a direct role in Zn^2+^ inflow [99,100].

Understanding the complex brain interactions that may lead to PD requires an understanding of how damage to the SNc impacts another area of the brain, i.e., the GABAergic rostromedial tegmental nucleus (RMTg) and how it changes the signaling of specific neurotransmitters. In the RMTg, especially AMPARs, which function as gates on brain cells, are essential for communication between neurons. The AMPARs in the RMTg became more sensitive and responsive after damage to the SNc, which is an area that is greatly impacted in PD. In SNc-damaged rats, the effects were more apparent and persistent, simulating Parkinsonian conditions, due to the increased sensitivity of these receptors to AMPA-like substances [101].

#### 4.1.3. Kainate Receptor (KAR)

The KAR is a specific type of ionotropic glutamate receptor involved in the modulation of synaptic transmission in the CNS. Although not as extensively studied as its AMPA and NMDA counterparts, the KAR plays a nuanced role in excitatory neurotransmission, synaptic plasticity, and neurotoxicity [46]. Recently, it has been shown that they appear at the presynapse to influence transmitter release on both excitatory and inhibitory synapses as well as at the postsynapse to mediate excitatory neurotransmission in numerous brain regions [102].

These receptors form the tetrameric protein; i.e., it is composed of four subunits. Each of these subunits carries a ligand-binding site and participates in the development of the ion channel. It was determined that there is only a small but considerable degree of homology between the fundamental structures of these proteins and other ionotropic glutamate receptor subtypes: about 40% homology to AMPAR and about 20% homology to NMDAR [103]. The tetrameric structure can be homomeric or heteromeric. The KAR subunits KA1, KA2, GluR5, GluR6, and GluR7 are the five recognized subunits. Each subunit can build functional receptors with characteristics. Each subunit is divided into four distinct regions: an extracellular ATD that controls the assembly and trafficking of the receptor; an LBD that binds the neurotransmitters (glutamate or kainate); a TMD made up of three transmembrane helices (M1, M3, and M4) and a re-entrant loop (M2) that changes in response to the stimulation, leading to the opening of the ion channel; and an intracellular region, which manages intracellular trafficking, signaling, and regulation. Thus, KAR’s intricate nature enables them to have a significant impact on neurotransmission in the brain [103,104,105,106,107].

##### Role of Kainate Receptor in Alzheimer’s Disease

KAR are found in synapses between mossy fibers and CA3 pyramidal cells [108]. Presynaptic and postsynaptic processes are used by these receptors to modulate neuronal networks. Presynaptic methods frequently entail neurotransmitter release modulation, whereas postsynaptic mechanisms involve direct effects on postsynaptic neurons, such as membrane potential modulation and downstream signaling pathways [109,110]. In one study, researchers discovered a decrease in GluK2 (old nomenclature, GluR6), a specific subunit of KARs, in the stratum lucidum of the CA3 area of an Alzheimer’s mice model (APP/PS1). This drop in GluK2 was associated with a reduction in the amplitude and decay duration of synaptic currents mediated by GluK2-containing KARs. This could imply that KARs, particularly GluK2-containing KARs, are less frequent or operate less effectively in AD. This could lead to decreased neural communication, contributing to the cognitive deterioration seen in AD. Interestingly, similar changes were seen in mice with presenilin or APP/APLP2 genetic deletions as well as organotypic cultures treated with -secretase inhibitors. Presenilin is a key component of the -secretase complex, which is involved in cleaving APP into its many components, including the Aβ peptides that form plaques in AD patients’ brains. The study discovered that the GluK2 protein can interact with both full-length and C-terminal APP fragments. This connection points to a possible mechanism for the observed changes: the APP may aid in the stabilization of KARs at synapses. As a result, any changes to the APP, such as those caused by genetic deletions or -secretase inhibitors, may impact the stability of KARs at synapses, resulting in the reported drop in GluK2 and subsequent changes in synaptic currents [111].

The hippocampus, a part of the brain important for learning and memory, contains neurons, and the systemic treatment of kainic acid (KA) induces a large rise in intracellular Ca^2+^, which results in the generation of ROS, mitochondrial malfunction, and apoptosis [112]. Additionally, KA causes the activation of glial cells, including astrocytes and microglia, which sets off inflammatory reactions typical of ND illnesses. Recent research has also shown that KA affects several intracellular processes, including the build-up of substances resembling lipofuscin, the induction of complement proteins, the processing of APP, and the expression of tau protein [113,114].

The study also showed how KA affects inflammasome-mediated signaling pathways, which may then cause signs of AD, such as neuronal degeneration, memory problems, and increased synthesis of Aβ protein [115,116]. It was observed that the NLRP3 inflammasome and the transcription factor nuclear factor kappa-light-chain-enhancer of activated B cells (NF-κB) were both activated when KA was added. Interleukin-1β (IL-1), a cytokine that promotes inflammation, and brain-derived neurotrophic factor (BDNF), a protein that protects neurons, were both produced because of this activation. The production of IL-1β and BDNF was lowered when either NLRP3 or NF-κB activity was inhibited with the drug Bay11-7082, indicating that the inflammation brought on by KA can be regulated by modifying these pathways. The research also discovered that NLRP3 and NF-κB activation were inhibited by cutting down the expression of two KARs, GluK1 (old nomenclature, GluR5) and Gluk3 (old nomenclature, GluR7). This suggests that the KA-triggered inflammatory response is mediated by KARs, which are upstream regulators of NLRP3 and NF-κB [116].

##### Role of Kainate Receptor in Parkinson’s Disease

Research on the involvement of KAR in PD is continuing, as KAR has not been investigated as much in relation to PD as other glutamate receptors, like NMDAR and AMPAR. Some studies have investigated the involvement of KAR, a component of the glutamate neurotransmitter system associated with PD, in excitotoxicity and brain cell death [74,117]. A particular form of PD known as autosomal recessive juvenile parkinsonism has been associated with the parkin protein, which is produced by the PARK2 gene. It has been reported that a subunit of the KAR known as GluK2 interacts with the parkin protein. Mutation in the PARK2 gene causes the accumulation of KAR on the synaptic membrane, and overactivity of this receptor leads to the excitotoxicity [118]. In further studies, it was revealed that by inhibiting the KAR with the specific antagonist UBP310, one can stop the loss of dopamine neurons, normalize aberrant neuron activity, and control a crucial KAR subunit. UBP310 also increased the survival of neurons in a specific brain region, the SNc, which is a part of the brain associated with the movement control, although these protective effects were not dependent on specific KAR (GluK1-GluK3) [119]. In an observation, it was seen that the KARs concentrated in the globus pallidus region of the brain govern some slow-moving signals even when other receptors like AMPARs were inhibited [120].

### 4.2. Metabotropic Glutamate Receptors

G-protein-coupled receptors (GPCRs) for the neurotransmitter glutamate are known as metabotropic glutamate receptors (mGluRs). Instead of creating ion channels like ionotropic glutamate receptors, mGluRs initiate intracellular signaling cascades. They have a “Venus’s flytrap” structure in the extracellular domain for glutamate binding, seven transmembrane domains structurally, and frequently form dimers [46]. mGluRs are divided into three categories, i.e., I, II, and III based on second messenger coupling, pharmacological characteristics, and sequence similarity. Group I receptors (mGluR1 and mGluR5) are connected to the Gq protein at the postsynaptic membrane and usually have an excitatory impact. As a result of their activation, phospholipase C is stimulated, which increases intracellular Ca^2+^ and activates PKC [121,122]. Group II (mGluR2 and mGluR3) receptors are connected to Gi/o proteins and are typically found on presynaptic neurons. Their activation blocks adenylate cyclase function, which lowers the generation of cyclic adenosine monophosphate (cAMP). This may result in a reduction in neurotransmitter release, acting as auto-receptors that give the presynaptic neuron negative feedback [46,123]. Group III receptors (mGluR4, mGluR6, mGluR7 and mGluR8), like Group II, are connected to the Gi/o protein and are frequently present on the presynaptic membrane. They both limit the release of neurotransmitters in a similar manner. Even though both group II and group III mGluRs have inhibitory effects and certain similarities in their general structure and signaling, they differ in terms of their distinct subtypes, localizations, functional roles, pharmacology, and implications in disease [124,125].

In the CNS, mGluRs play a critical role in modulating glutamatergic neurotransmission. mGluRs participate in slower, longer-lasting signaling activities than their ionotropic cousins, which mediate rapid synaptic transmission. They function by turning on G proteins, which then set off multiple intracellular signaling pathways and affect a variety of cellular processes. The regulation of synaptic strength and plasticity, management of neuronal excitability, and manipulation of the release of different neurotransmitters are all functions of mGluRs. In addition to controlling neuronal excitability and modulating the release of different neurotransmitters, mGluRs also regulate synaptic strength and plasticity. The roles mGluRs play in memory, anxiety, learning, and pain perception are influenced by these processes. Additionally, they play a key role in controlling the brain’s excitatory and inhibitory balance, which makes them essential for preserving neuronal homeostasis [3,124].

#### 4.2.1. Role of Metabotropic Receptor in Alzheimer’s Disease

In the pathophysiology of AD, mGluRs are now recognized as important players. Group I mGluRs, which include mGluR1 and mGluR5, have a complex and important role in AD. These receptors participate in processes that are both neuroprotective and neurodegeneration that have a complex function in the control of synaptic activity [3,124]. mGluR5 plays a role in the pathogenesis of AD by interacting with Aβ oligomers (Aβos) and tau protein. mGluR5 has been demonstrated to be essential in mediating Aβo toxicity in preclinical AD models. It does this through several methods, including promoting the clustering of Aβo and serving as a co-receptor for Aβos that binds to the cellular prion protein (PrPc), activating the tyrosine kinase Fyn. The tau protein is phosphorylated because of this activation’s secondary effects. It is interesting to note that mGluR5 may also connect the Aβ and tau pathologies in AD. When Aβo and PrPc form complexes, mGluR5 is recruited, which further activates Fyn and causes tau to be phosphorylated. Furthermore, functional tau is necessary for the postsynaptic targeting of Fyn, leading to excitotoxicity mediated by NMDARs [126,127,128].

A specific AD mouse model (APPswe/PS1E9) was used in the study, and it was found that the genetic deletion of mGluR5 can improve spatial learning deficiencies. This deletion also decreases the development of Aβ oligomers and the quantity of plaques, proving that mGluR5 expression is a factor in Aβ accumulation. The study also discovered elevated mammalian target of rapamycin (mTOR) phosphorylation and fragile X mental retardation protein expression in the APPswe/PS1E9 mice. These alterations were similarly diminished by the removal of mGluR5. The study draws the conclusion that mGluR5 activation by Aβ seems to set off a positive feedback loop that aggravates Aβ formation and AD pathogenesis [129].

It has been discovered that mGluRs play a crucial role in the processing of synaptic APP. Studies using isolated nerve terminals from TgCRND8 mice, which express human APP with familial AD mutations, have revealed captivating connections between glutamatergic neurotransmission and Aβ production [130,131,132]. It is noteworthy that activation with the group II mGluR agonist DCG-IV resulted in a rise in APP C-terminal fragments, which was followed by a phase of breakdown and a prolonged accumulation of Aβ42. A group II mGluR antagonist, LY341495, inhibited this procedure, demonstrating that group II mGluRs may activate all three secretases necessary for APP processing [133].

One of the studies looked at the connection between mGluRs and AD, concentrating on how the Aβ affected prefrontal cortical pyramidal neurons. The activation of group I mGluRs by a particular agonist increased inhibitory postsynaptic current amplitude under basal conditions, and this mechanism was found to be PKC-sensitive. NMDAR currents were also enhanced by group II mGluR activation through a PKC-dependent mechanism. These alterations, however, were eliminated when Aβ was included, proving that Aβ can obstruct mGluRs from performing their normal role. Additionally, both group I and group II mGluR-mediated activation of PKC was inhibited in the presence of Aβ. This study has profound implications for our knowledge of the pathogenesis of AD, since it raises the possibility that Aβ may impair cognitive function by interfering with normal mGluR activity, notably the control of GABA transmission and NMDAR currents by mGluR [134,135,136].

The study has also examined the function of group III mGluR receptors in the brain’s immune cells called microglia with a particular focus on their potential connection to ND conditions like AD. The study found that activating group III mGluR receptors with agonists like (L)-2-amino-4-phosphono-butyric acid or (R, S)-phosphonophenylglycine decreased the synthesis of cAMP, which is connected to a negative inhibition of adenylate cyclase. It is significant because activating these receptors did not seem to harm microglia [137].

#### 4.2.2. Role of Metabotropic Glutamate Receptor in Parkinson’s Disease

Group I mGluRs that include mGluR1 and mGluR5 are mostly concentrated at the postsynaptic site of neurons, but they are also reported to be present in the presynaptic sites of nigrostriatal dopamine terminals, which is a region linked to movement [138]. It has been revealed that the expression of mGluR5 is increased in the dopaminergic brain region of PD, and this mGluR5 is also known to interact and boost the activity of ionotropic NR2B containing NMDAR by phosphorylating the NR2B subunit, leading to Ca^2+^ imbalance and neurotoxicity [117,139,140].

The microglia, a type of immune cell in the brain, are activated by a protein called ⍺-synnuclein, which is a crucial stage in the pathogenesis of PD. This stimulation damages brain cells by causing inflammation and neurotoxicity [141]. One study revealed how mGluR5 interacts with ⍺-synuclein and how it may control these negative consequences. It was reported that mGluR5 upon binding to ⍺-synuclein at its N-terminal became activated and protected against neurotoxicity by reducing the inflammation and cytokine production brought on by ⍺-synuclein. However, mGluR5 was discovered to be co-located in lysosomes with ⍺-synuclein, suggesting that the overexpression of ⍺-synuclein caused a decrease in mGluR5 through a lysosomal degradation mechanism. Both in vitro cell cultures and in vivo rat models of the disease supported these extensive relationships, suggesting a potential pathway that could be targeted for neuroprotective therapy in PD [142].

Group II mGluRs are located at the terminals coming from the subthalamic nucleus and entering the substantia nigra pars reticulata (SNr); when activated, they decrease the excitatory activity at the SNr synapses. Therefore, it is predicted that increasing the activity of their receptors can offer more adaptable therapeutic strategy [143].

Group III mGluRs, and more especially mGluR4, which is concentrated in the vital area of the brain called the globus pallidus (GP), were studied to see how they alter the communication between brain regions that govern movement control [144]. It is known that in PD, a lot of inhibitory signals are sent to the GP that affects movement control. mGluR4 reduces excessive inhibition in GP, leading to the idea that activating mGluR4 can help control this movement disorder [139,145].

## 5. Platelets as a Cellular Model for Understanding Neuronal Mechanism and Neurological Disorders

Because platelets and neurons both originate from the ectoderm and exhibit functional and anatomical similarities, platelets are a great model system for studying intricate neural processes [146,147,148]. For instance, both platelets and neurons have a comparable serotonin transport pathway [149]. Beyond serotonin, these two cell types also release other significant neurotransmitters like dopamine, gamma-aminobutyric acid (GABA), and glutamate [146]. In addition, glutamate receptors, which are essential for the proper operation of the nervous system, are found in non-neuronal tissues such bone marrow, the kidney, the heart, and the pancreas in addition to neurons and platelets [150]. Platelets have proven crucial in shedding light on numerous neurological illnesses like migraines [151], PD [152], autism [153], schizophrenia [154], and AD [155] due to these similar properties. Research on AD has revealed that platelets are the main source of circulating APP and there is increased activity of platelet β-secretase activity—a key enzyme involved in the production of amyloid-β (Aβ) peptides—which are essential for the development of the Aβ plaques that are a hallmark of the condition [20]. We demonstrated previously that the prion protein, also expressed in platelets and bearing a beta-amyloid structure like the Aβ protein linked with AD, can when misfolded into its scrapie form trigger notable platelet activation and cytotoxicity, highlighting the potential dangers of protein misfolding and aggregation in cellular environments [156,157]. Furthermore, platelets exhibit increased adhesion to the subendothelial matrix component, and they have been observed in cerebral blood vessels, emphasizing their potential interaction with the cerebral vasculature. Importantly, both APP and amyloid-β are implicated in influencing platelet function. Platelets also play a role in enhancing the formation of amyloid-β aggregates within cerebral vessels, implicating them in the pathology of cerebral amyloid angiopathy (CAA). Additionally, platelets seem to promote neuroinflammation and vessel damage in the context of AD. These findings collectively underscore the intricate relationship between platelets and AD, with platelets influencing and being influenced by Aβ accumulation, neuroinflammation, and vascular dysfunction in the disease’s pathogenesis. In the context of PD, platelets exhibit notable alterations that provide valuable insights into the disease’s pathophysiology. These changes include an increased mean platelet volume [158] as well as a decrease in vesicular monoamine transporter 2 mRNA in human platelets, which is a critical component of dopamine transport, showing a link between platelets and the neurotransmitter systems affected in PD [159]. Moreover, platelet mitochondrial dysfunction is observed, indicating potential issues with cellular energy production [160,161,162]. Earlier studies have reported that platelets play a pivotal role in connecting the peripheral metabolic and inflammatory mechanisms with central nervous system processes. In metabolic syndrome, which is characterized by a cluster of conditions like obesity, high blood pressure, and insulin resistance, platelets become activated. This activation can have significant effects on neuronal processes and contribute to the development and progression of neurodegenerative diseases [163]. These findings collectively underscore the significance of platelets in AD and PD, as they appear to be influenced by and potentially contribute to the neurodegenerative processes associated with the disease.

### The Role of Platelet’s Glutamate and Its Transporter in Alzheimer’s and Parkinson’s Disease

Although there has been much research on the central nervous system’s glutamate receptors in relation to NDDs, the role of platelets in regulating glutamate levels—and potentially having an impact on these diseases—is a relatively recent field of study. Platelets, generally known for their function in hemostasis [164], also play a significant part in the control of neurotransmitters. We and others have earlier demonstrated that glutamate, stored within the dense granules of platelets, is released upon activation and plays a pivotal role in promoting platelet activation [44]. Platelet has glutamate transporters, i.e., excitatory amino acid transporter 1, 2 and 3 (EAAT1, EAAT2 and EAAT3) that are comparable to those in the brain and are responsible for removing glutamate from the bloodstream [165]. One such study has revealed some interesting age-related changes in glutamate uptake by platelets (Figure 6). The maximal rate of glutamate absorption (Vmax) declines with age as a natural feature of aging. Notably, this decreased Vmax seems to be linked to a drop in the EAAT1 glutamate transporter’s expression levels. This age-related decline becomes considerably severe in the context of AD. EAAT1 levels are significantly lower in AD patients compared to controls of similar age. A possible explanation for the observed deficit in glutamate uptake in platelets from AD patients could be provided by this extra decrease in transporter expression [166]. Like this, the severity of the symptoms of PD is also connected with a significant decline in platelet efficiency regarding the absorption of glutamate. This suggests that platelets are less effective in AD and PD at acting as “clean-up” cells for excess glutamate in the circulation [167].

One of the studies delves into the behavior of glutamate uptake in human platelets and its potential relevance to neurological disorders. The study reveals that when platelets were incubated with [3H] glutamate, there was a dose-dependent increase in glutamate uptake with maximal velocity rate enhancement but no alteration in affinity for glutamate. Glutamate receptor agonists and the molecules like glutamate did not mimic this effect, suggesting a specific response. The research further investigates the impact on major glutamate transporters, highlighting a significant increase in EAAT1 expression following glutamate preincubation, which is an effect blocked by cycloheximide. Moreover, the involvement of EAAT2 is suggested with complete abolishment of the glutamate stimulation effect when EAAT2 is selectively inhibited. These findings emphasize the intricate substrate-dependent modulation of glutamate uptake in human platelets. These results indicate that platelets can serve as a valuable model for investigating the dysregulation of glutamate uptake in neurological disorders, offering insights into potential mechanisms underlying these conditions [165].

One of the studies co-cultured primary neuronal cultures and activated platelets to evaluate the effect of activated platelets on neuronal health and functions. This study posited that thrombin-activated platelets release significant amounts of glutamate with concentrations reaching beyond 300 μM, and when such activated platelets were introduced to neuronal cultures, they had a neurotoxic effect. Interestingly, the neurotoxicity was somewhat mitigated using glutamate receptor antagonists, suggesting the role of glutamate in this process. The study also demonstrated that exposure to these activated platelets led to a substantial downregulation in the surface presence of the glutamate AMPAR subunit GluR2, which is generally considered an indicator of excitotoxic exposure and might be a mechanism contributing to neuronal dysfunction [168,169].

## 6. Current Pharmacological Treatments for Alzheimer’s and Parkinson’s Disease

As we explore the complexity of NDDs like AD and PD, it is crucial to not only focus on the emerging research but also understand the current pharmacological options that are available for patient care. There is an exhaustive list of medications commonly prescribed in the clinical management of both AD (Table 1) and PD (Table 2) as approved by the Food and Drug Administration (FDA). These medications primarily aim to alleviate symptoms by acting on various neurotransmitter systems in the brain, including the glutamatergic system, which is increasingly recognized for its role in these conditions. While these drugs can often provide temporary relief of symptoms and sometimes slow the rate of cognitive or motor decline, they are unfortunately not curative and do not significantly alter the disease progression. Thus, these medications, although important, serve more as palliative treatments rather than definitive solutions [170,171,172,173]. This current state of therapeutic options emphasizes the need for more research into targeted treatments, particularly as we expand our understanding of the role that glutamate receptors play in the pathophysiology of AD and PD.

## 7. Conclusions

This article has reviewed the complex role played by glutamate receptors in AD and PD. Glutamatergic signaling dysregulation is a key factor in the pathophysiology of both disorders even though the specific mechanics may differ. The illness development is neither reversed nor stopped by current drugs, which only partially target these pathways yet provide symptomatic relief. To do more than just treat symptoms, future research should concentrate on creating targeted medicines that more effectively address glutamatergic dysfunction. New opportunities for therapeutic intervention in these complicated NDDs are likely to open as our understanding of glutamate receptors continues to grow. 

## Figures and Tables

**Figure 1 biomolecules-13-01609-f001:**
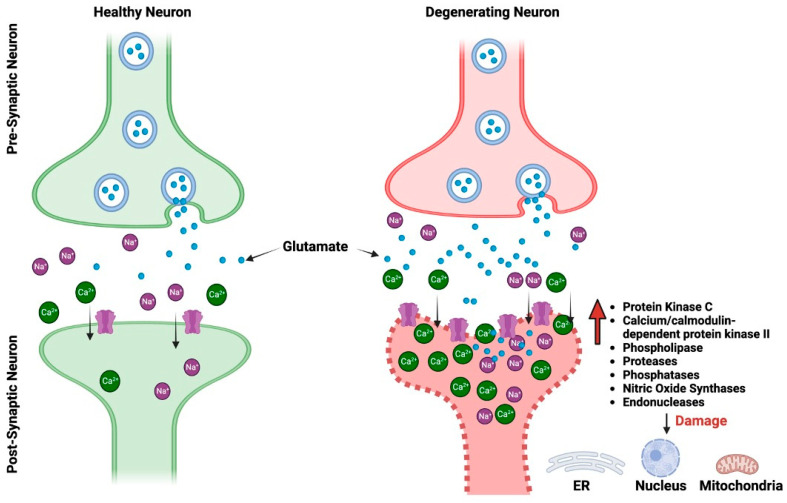
The pictorial comparison between healthy neurons and neurons impacted by glutamate excitotoxicity. ER, endoplasmic reticulum.

**Figure 2 biomolecules-13-01609-f002:**
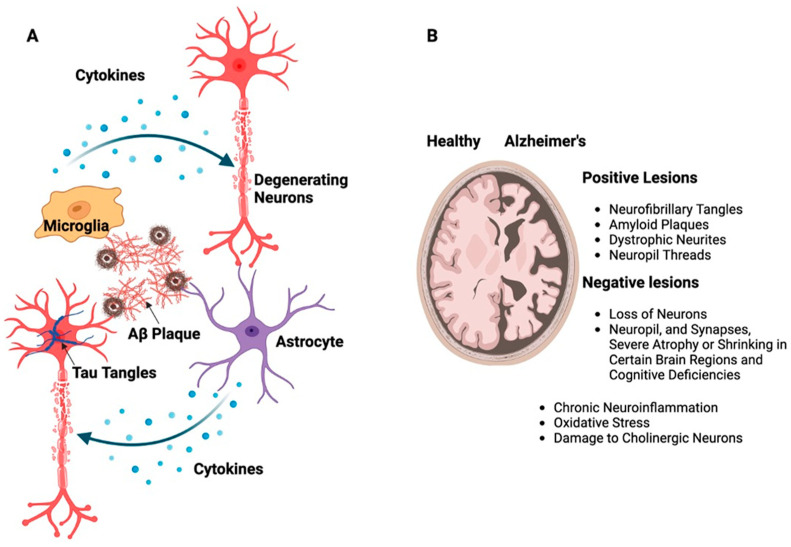
(**A**) Schematic representation of building up Aβ and tau proteins leading to the formation of large, insoluble amyloid fibrils and NFTs. These structures contribute to the development of plaques as well as soluble oligomers that disperse throughout the brain resulting in neurotoxicity, which, in turn, activates supporting brain cells like astrocytes and microglia. (**B**) The illustration shows the difference between a healthy brain and a brain affected by AD, emphasizing the presence of neurotic plaques and NFTs, particularly in the medial temporal lobe and neocortical areas of the brain.

**Figure 3 biomolecules-13-01609-f003:**
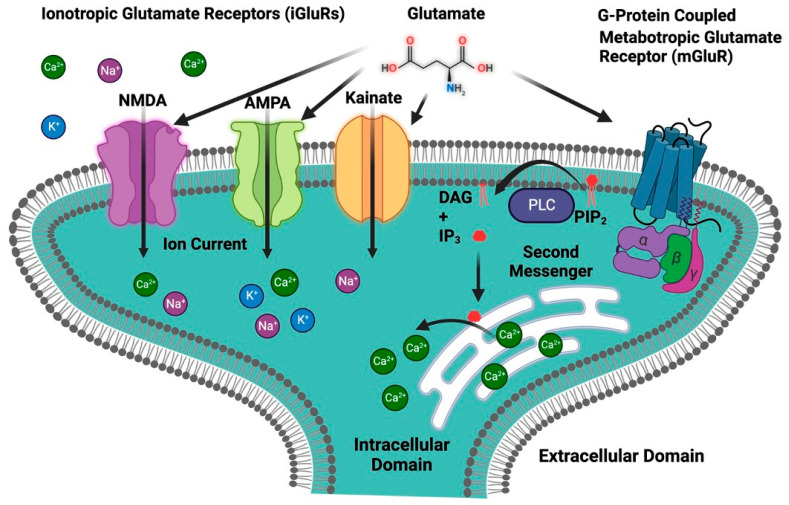
Classification of glutamate receptors. The first group is ionotropic glutamate receptors (iGluRs), which include an ion channel and are further divided into three primary subtypes: NMDA, AMPA, and kainate receptors. The second group is metabotropic glutamate receptors (mGluRs), which are G-protein-coupled receptors. IP_3_, inositol 1,4,5-trisphosphate; DAG, diacylglycerol; PIP_2_, phosphatidylinositol 4,5-bisphosphate; PLC, phospholipase C.

**Figure 4 biomolecules-13-01609-f004:**
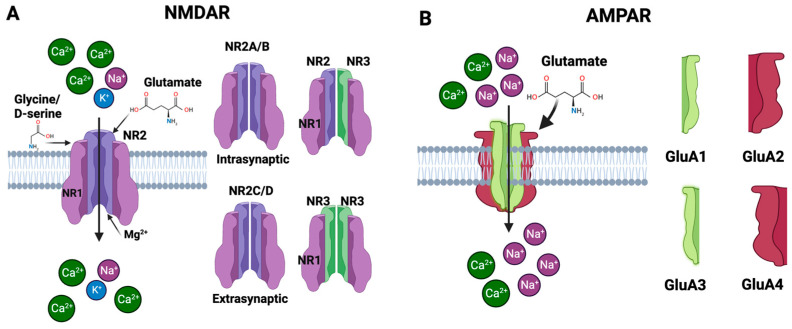
Subunits of the NMDAR and AMPAR. (**A**) The NMDAR is primarily composed of NR1 and NR2 subunits that bind to glycine and glutamate, respectively. A functional receptor can also be created by joining NR1 and NR2 subunits with a third type of subunit, NR3. The NMDAR has special permeability to Ca^2+^. (**B**) The AMPAR is a tetrameric structure composed of four subunits—GluA1, GluA2, GluA3, and GluA4. When glutamate binds to the receptor, it triggers the ion channel to open, allowing primarily Na^+^ and, under certain conditions, Ca^2+^ to enter the cell.

**Figure 5 biomolecules-13-01609-f005:**
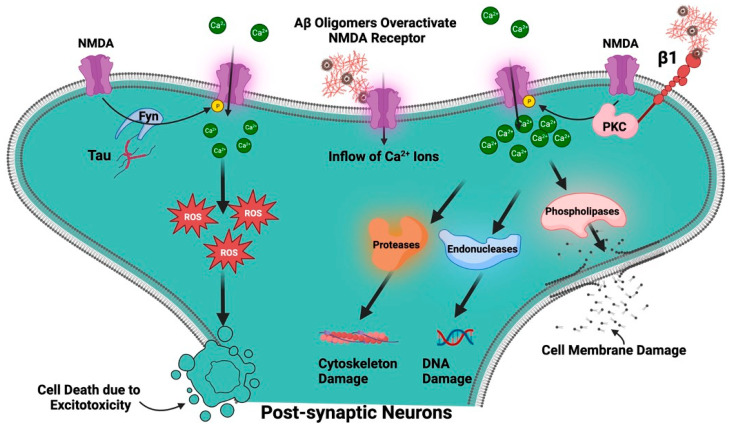
Role of NMDAR in AD. Aβ oligomers overactivate NMDARs, which causes a significant inflow of Ca^2+^ ions into the cell associated with generation of ROS. Aβ oligomers activate integrin β1 and protein kinase C (PKC), which, in turn increases the conductance of the NMDAR, leading to a boost in Ca^2+^ influx and excitotoxicity. The tyrosine kinase, Fyn, phosphorylates NMDAR. Tau appears to function as a scaffold that brings Fyn and NMDARs into proximity, thus facilitating phosphorylation of the receptor.

**Figure 6 biomolecules-13-01609-f006:**
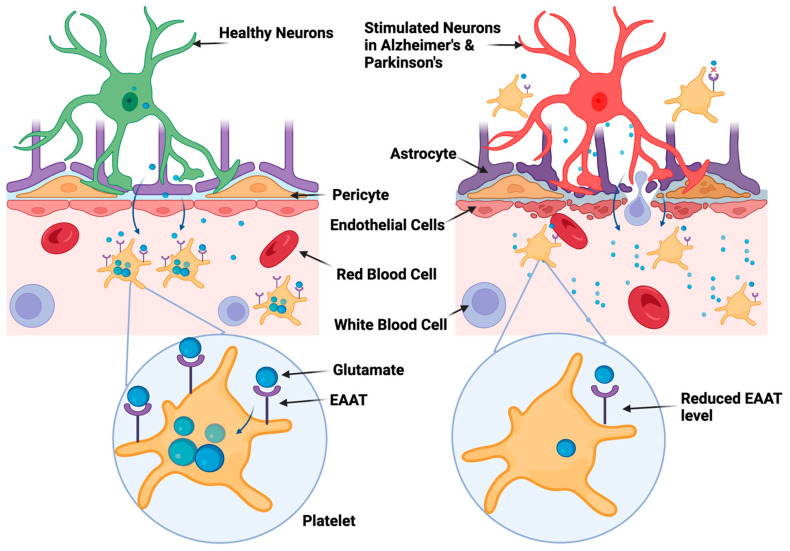
Schematic representation of impairment of glutamate uptake in AD and PD. Elevated levels of glutamate in the bloodstream are usually cleared by platelets through the excitatory amino acid transporter (EAAT) receptor, helping to maintain its normal concentration in the blood. However, when the blood–brain barrier is compromised in AD and PD, platelets could cross into the central nervous system. Moreover, the glutamate uptake is impaired not just in the brain but also in platelets. This may contribute to the elevated levels of glutamate that have been observed in AD and PD, which in turn could exacerbate excitotoxicity and neuronal death.

**Table 1 biomolecules-13-01609-t001:** FDA-approved pharmaceutical agents used for clinical management of Alzheimer’s disease.

Drug Name	Drug Class	Mode of Action	Approved Indications	Common Side Effects	Manufacturer and Year of Approval
Donepezil	Acetylcholinest-erase inhibitor	Increases acetylcholine levels in the brain by inhibiting its breakdown	Mild to severe AD	Nausea, diarrhea, insomnia, vomiting, muscle cramps, fatigue	Eisai Co. (brand name: Aricept), 1996
Rivastigmine	Acetylcholinest-erase inhibitor	Increase acetycholine and butyrylcholine levels in the brain by inhibiting their breakdown	Mild to moderate AD and PD dementia	Nausea, vomiting, loss of appetite, dizziness, weight loss	Novartis AG (brand name: Exelon), 2000
Galantamine	Acetylcholinest-erase inhibitor	Increases acetylcholine levels by inhibiting its breakdown and modulating nicotinic acetylcholine receptors	Mild to moderate AD	Nausea, vomiting, diarrhea, weight loss, loss of appetite	Janssen Pharmaceuticals (brand name: Razadyne), 2001
Memantine (Namenda)	NMDAR antagonist	Modulate the activity of glutamate, an excitatory neurotransmitter, by blocking NMDAR	Moderate to severe AD	Dizziness, confusion, headache, constipation, cough	Allergan (brand name: Namenda), 2003
Memantine+ Donepezil	Combination of NMDAR antagonist and acetylcholinesterase inhibitor	Combines the actions of Memantine andDonepezil	Moderate to severe AD	Headache, diarrhea, dizziness, flu symptoms, cough	Allergan and Eisai Co. (brand name: Namzaric), 2014
Leqembi (Lecanemab)	IgG1 monoclonal antibody	Binds with high affinity to Aβ-soluble protofibrils and helps break them down	Early stage of AD	Amyloid-related imaging abnormalities (ARIA), a temporary swelling and/or bleeding in certain areas of the brain that usually resolves with time. ARIA risk may increase in those who have two copies of the well-known Alzheimer’s risk gene	Eisai/Biogen, 2023
Donenmab	Monoclonal Ab, anti-Aβ	Binds with high affinity to Aβ-soluble protofibrils and helps break them down	Early stage of AD	Headaches, reactions to intravenous drip, swelling and microbleed in the brain	Eli Lilly (it is in the large final stage trial, called TRAILBLAIZER-ALZ2)

**Table 2 biomolecules-13-01609-t002:** FDA-approved pharmaceutical agents used for clinical management of Parkinson’s disease.

Drug Name	Drug Class	Mode of Action	Approved Indications	Common Side Effects	Manufacturer and Year of Approval
Levodopa/Carbidopa (Sinemet)	Dopamine precursor	Increases brain levels of dopamine. Carbidopa prevents peripheral breakdown of levodopa	PD, parkinsonism	Nausea, vomiting, dyskinesia, orthostatic hypotension	Merck Sharp and Dohme, 1975
Pramipexole (Mirapex)	Dopamine agonist	Dopamine agonist: Mimics the effects dopamine in the brain	PD, restless leg syndrome	Nausea, vomiting, loss of appetite, dizziness, weight loss.	Boehringer Ingelheim, 1997
Ropinirole (Requip)	Dopamine agonist	Dopamine agonist: Mimics the effects dopamine in the brain	PD, restless leg syndrome	Fatigue, dizziness, hallucination, nausea	GlaxoSmithKline, 1998
Entacapone (Comtan)	COMT inhibitor	COMT inhibitor prolongs the effects of levodopa by blocking its breakdown in the periphery	PD (adjunct to levodopa/carbidopa)	Nausea, abdominal pain, diarrhea, discoloration of urine	Orion Pharma, 1999
Apomorphine (Apokyn)	Dopamine agonist	Dopamine agonist: Rapidly treats off-episodes by mimicking dopamine	PD (far off episodes)	Yawning, dyskinesias, dizziness, rhinorrhea	Britannia pharmaceuticals, 2004
Rasagiline (Azilect)	MAO-B inhibitor	MAO-B inhibitor: blocks the breakdown of dopamine in the brain	PD	Joint pain, depression, dyskinesias, flu-like symptoms.	Teva pharmaceuticals, 2006
Safinamide (Xadago)	MAO-B inhibitor and glutamate release inhibitor	MAO-B inhibitor and glutamate release inhibitor: enhances dopaminergic activity and reduces glutamate	PD (add-on treatment to levodopa/ carbidopa)	Dyskinesia, hypertension, hallucinations, falls.	Zambon, 2017

COMT: catechol-O-methyl transferase, MAO-B: monoamine oxidase type B.

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
