# Peer review of "Glutamate Receptor Dysregulation and Platelet Glutamate Dynamics in Alzheimer’s and Parkinson’s Diseases: Insights into Current Medications"

_biomolecules, 2023, doi:10.3390/biom13111609_

Round 1
Reviewer 1 Report
Comments and Suggestions for Authors
In this review Gautam et. al., reviews the current medications and glutamate receptor antagonists for AD and PD, with their mechanism of action. Authors tried to focus on the involvement of glutamate receptor and its systemic regulation by platelets could be useful for effective treatments, delaying disease progression.
Overall this review holds a great potential to answer a clinically significant question. Furthermore, carries up-to-date information in the field of neuro-degeneration, with a novel proposal, highlighting the involvement and role of platelets regulating AD & PD progression and possible line of treatment.
Minor comments:
1. Authors need to rewrite the abstract focusing more about ‘role of platelets in AD & PD disease progression’”.
2. It would be good if authors can shed some light on platelet activation in metabolic syndrome and neuro-degeneration and possible treatments.
Reviewer 2 Report
Comments and Suggestions for Authors
This is a well-structured and comprehensible review on the role of glutamatergic receptors in Alzheimer’s and Parkinson’s diseases. Gautam et al. address that these glutamatergic receptors may be an important mechanism to control neuronal activity as well as in platelets. In addition, the authors address several medication strategies for Alzheimer’s and Parkinson’s diseases. This review is well written and organized in clear sections. The manuscript makes a remarkable review of the literature, including recent discoveries on the field. I have only one suggestion that could be included in the manuscript:
Would it be possible to study glutamatergic activity in platelets as a means of prognosis of the disease? This could be a diagnostic method using a very non-invasive technique.
Reviewer 3 Report
Comments and Suggestions for Authors
Presented is the detailed review of the role of glutamatergic signaling and specifically different classes of glutamate receptors in mechanisms of development of some common NDDs, such as AD and PD. Also, the platelets are presented as a useful model of glutamate dynamics, providing the other source for understanding the pathophysiology of AD and PD. The manuscript is well written and structured, and easy to follow.
Minor comments:
line 78: account for -- accounts?
line 178: for a total of 5.8 million years in 2019 - years or people? This is already said in line 176, so either can be removed
Figure 6: Abbreviations for RBC and WBC should be deciphered.
Comments on the Quality of English Language
English is of an appropriate quality although the minor editing can be useful.
Reviewer 4 Report
Comments and Suggestions for Authors
This is an excellent review article considering the role of glutamate receptors in the context of the 2 major neurodegenerative diseases - AD and PD. Additionally, the authors present accumulated evidence for the roles of platelets in glutamate secretion and reuptake in the context of neurodegeneration. The article is particularly nicelly illustrated with broad literature consideration and summary of therapeutic approches concerning glutamate metabolism. As it is really of a textbook scope and lenghth I would suggest to authors to shorten the manuscript, particularly parts descriging general features of AD and PD.
